# Differential contributions of the two human cerebral hemispheres to action timing

Anja Pflug[1†], Florian Gompf[1†], Muthuraman Muthuraman[2], Sergiu Groppa[2], Christian Alexander Kell[1]*

[1]Cognitive Neuroscience Group, Brain Imaging Center and Department of Neurology, Goethe University, Frankfurt, Germany; [2]Movement Disorders and Neurostimulation, Biomedical Statistics and Multimodal Signal Processing Unit, Department of Neurology, Johannes Gutenberg University, Mainz, Germany

**Abstract** Rhythmic actions benefit from synchronization with external events. Auditory-paced finger tapping studies indicate the two cerebral hemispheres preferentially control different rhythms. It is unclear whether left-lateralized processing of faster rhythms and right-lateralized processing of slower rhythms bases upon hemispheric timing differences that arise in the motor or sensory system or whether asymmetry results from lateralized sensorimotor interactions. We measured fMRI and MEG during symmetric finger tapping, in which fast tapping was defined as auditory-motor synchronization at 2.5 Hz. Slow tapping corresponded to tapping to every fourth auditory beat (0.625 Hz). We demonstrate that the left auditory cortex preferentially represents the relative fast rhythm in an amplitude modulation of low beta oscillations while the right auditory cortex additionally represents the internally generated slower rhythm. We show coupling of auditory-motor beta oscillations supports building a metric structure. Our findings reveal a strong contribution of sensory cortices to hemispheric specialization in action control.
DOI: https://doi.org/10.7554/eLife.48404.001

*For correspondence:
c.kell@em.uni-frankfurt.de

[†]These authors contributed equally to this work

Competing interests: The authors declare that no competing interests exist.

## Introduction

Functional asymmetries between the two hemispheres are an intriguing principle of brain organization. On the behavioral level, these become most evident in the way humans use their hands. In tasks requiring movements of both hands, right-handers typically use the right hand for the faster, dynamic movements while the left hand is used for slower movements, or even static control of hand position (*Sainburg, 2002*; *Swinnen and Wenderoth, 2004*; *Serrien and Sovijärvi-Spapé, 2015*). Cutting bread or hammering a nail into the wall represents everyday examples for such functional asymmetries. In the lab, finger tapping can be used to detect hemispheric asymmetries related to this phenomenon. Typically, the right hand taps relative higher tapping frequencies more precisely than the left hand even in bimanual monofrequent finger tapping (*Repp, 2005*; *Ivry, 1996*; *Peters, 1980*). Conversely, the left hand taps relative lower tapping frequencies more precisely than the right hand (*Pflug et al., 2017*). This suggests the left hemisphere preferentially controls relative higher tapping frequencies and the right hemisphere preferentially controls relative lower tapping frequencies, but the origins of such hemispheric asymmetries are not known. There are several proposals on the origins of functional differences between the hemispheres ranging from specialized processing in the sensory domain, lateralized sensorimotor interactions, asymmetric motor control, to domain-general frameworks on hemispheric dominance (*Kimura, 1993*; *Minagawa-Kawai et al., 2007*; *Toga and Thompson, 2003*; *Kell and Keller, 2016*). Behavior could benefit from parallel processing of different aspects of complex stimuli and/or movement planning in the left and right

**eLife digest** If you watch a skilled pianist, you will see that their right hand moves quickly up and down the high notes while their left hand plays the lower notes more slowly. Given that about 90% of people are right-handed, it might not seem too surprising that most people can perform faster and more precise movements with their right hand than their left. Indeed, when right-handers perform any task, from hammering a nail to slicing bread, they use their right hand for the faster and more difficult action, and their left hand for slower or stabilizing actions.

But why? It could be that the left hand is simply less capable of performing skilled movements than the right. But another possibility is that the left hand is actually better than the right hand when it comes to slower movements. To test this idea, Pflug, Gompf et al. asked healthy volunteers to tap along to a metronome with both index fingers. On some trials, the volunteers had to tap along to every beat. On others, they had to tap in time with every fourth beat. While the volunteers performed the task, Pflug, Gompf et al. measured their brain activity.

The results showed that the volunteers, who were all right-handed, followed the fast rhythm more precisely with their right hand than with their left. But they tapped the slow rhythm more accurately with their left hand than with their right. Areas of the brain that process sounds showed increased activity during the task. This increase was greater on the left side of the brain – which controls movement of the right side of the body – when the volunteers tracked the faster rhythm. By contrast, sound-processing areas on the right side of the brain – which controls the left side of the body – showed greater activity when participants tapped the slow rhythm.

The findings thus suggest that the left half of the brain is better at controlling faster rates of movement, whereas the right half is better at controlling movements with slower rhythms. This could also help explain why in most people the left side of the brain controls speech. Speech requires rapid movements of the lips, tongue and jaw, and so it may be better controlled by the left hemisphere. Understanding how the two hemispheres control different actions could ultimately lead to new strategies for restoring skills lost as a result of brain injuries such as stroke.

DOI: https://doi.org/10.7554/eLife.48404.002

hemisphere (*Serrien et al., 2006*). Influential theories suggest differential sensory processing of relative frequencies either in the spectral or the temporal domain (*Ivry and Robertson, 1998*; *Flevaris and Robertson, 2016*; *Poeppel, 2003*) as computational bases of hemispheric specialization. However, empirical studies in which spectral or temporal aspects of the sensory input were parameterized did not always support those theories (*Luo et al., 2007*; *Giraud and Truy, 2002*; *Boemio et al., 2005*). This could represent a consequence of the fact that brain activity is only subtly lateralized during perceptual tasks. However, functional lateralization is thought to be amplified once a motor output is required (*Ivry and Robertson, 1998*; *Keller and Kell, 2016*).

To dissociate the specific contributions of the sensory and motor systems to functional lateralization of hand control, we performed two imaging studies using functional magnetic resonance imaging (fMRI) and magnetoencephalography (MEG). The study design excluded that condition effects resulted from sensory stimulus features or differential effector use. In an auditory-paced finger tapping paradigm, participants were asked to tap bimanually to auditory beats. Tapping to every auditory beat (2.5 Hz) was defined as the fast tapping condition while tapping to every fourth auditory beat (beat position four) represented slow tapping at 0.625 Hz (see *Figure 1*). Both frequencies fall into the natural range of finger movements but represent different ends of the spectrum (*Parncutt and Cohen, 1995*; *London, 2012*; *Drake and Palmer, 2000*; *Repp, 2003*). While in the fast tapping condition, the fast auditory beat was the only rhythm that was processed and used for auditory-motor synchronization, this faster rhythm served as a timing signal to generate a slower rhythm in the slow tapping condition. The slow tapping condition was of primary interest in our study, because during slow tapping two interrelated rhythms had to be represented in parallel, a condition that could potentially reveal hemispheric specialization for controlling rhythms of different relative frequencies (*Ivry and Robertson, 1998*). A prior behavioral study (*Pflug et al., 2017*) suggested that representing a relative slow rhythm in parallel to a faster one should reveal the contribution of the right hemisphere to hand control. While we used fMRI to detect whether auditory or

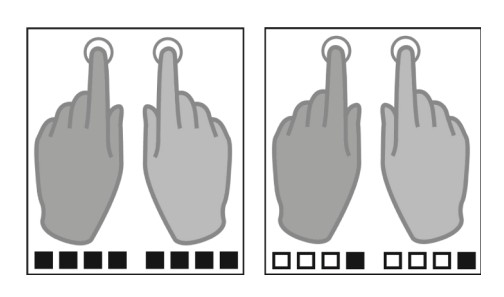

**Figure 1.** Tapping conditions. Participants were instructed to tap either to every beat (fast tapping, left panel) or to the fourth position of four consequent auditory beats (slow condition, right panel). Filled squares represent tapping events, white squares represent auditory beats without tapping in the slow tapping condition.
DOI: https://doi.org/10.7554/eLife.48404.003

motor regions show a more pronounced lateralization profile, which answers the question of different contributions of the sensory and motor systems to hemispheric specialization, we used MEG to identify hemispheric differences in brain rhythms associated with finger tapping in a time-resolved manner and to investigate time resolved directed connectivity between auditory and motor association cortices (*Figure 2*).

Movement is known to suppress beta oscillations and to increase activity in the gamma range (*Muthuraman et al., 2012*; *Tamás et al., 2018*; *Pfurtscheller et al., 2003*; *Pfurtscheller and Lopes da Silva, 1999*; *Engel and Fries, 2010*). Yet, neural oscillations are, particularly in the beta range, not only reflected current motor state but also implicated in internal timing, especially during rhythm processing, and are amplitude-modulated during rhythm perception and production not only in the motor and supplementary motor cortex, but also in the auditory and auditory association cortex (*Arnal and Giraud, 2012*; *Doelling and Poeppel, 2015*; *Nobre et al., 2007*; *Fujioka et al., 2015*; *Meijer et al., 2016*; *Morillon et al., 2014*; *Kilavik et al., 2013*; *Kulashekhar et al., 2016*; *Morillon and Baillet, 2017*; *Iversen et al., 2009*). Comparing neural oscillations during slow and fast rhythmic finger tapping may reveal the way the brain represents the two different rhythms in parallel. Amplitude modulations of beta oscillations should differ between functional homologues in case there were hemispheric processing differences in timing of relative tapping frequencies. We hypothesized that motor and/or auditory cortices may not only differ in overall beta power but also in terms of their degree of representing the slow and fast rhythms in the temporal modulation of beta power (*Fujioka et al., 2015*; *Morillon and Baillet, 2017*). If the predictions from the signal-driven hypotheses on hemispheric specialization (*Ivry and Robertson, 1998*) hold true, we specifically expected the right auditory cortex to more strongly represent the slow rhythm and the left auditory cortex the fast rhythm during slow finger tapping, the condition that comprised both rhythms. A left dominance in hand motor control based on left-lateralized sequencing skills

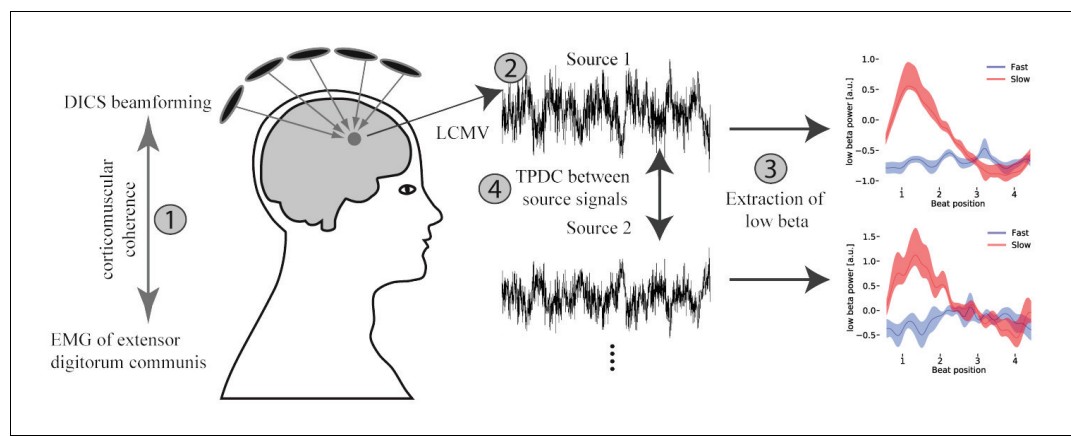

**Figure 2.** Overview of the applied MEG analysis steps. (1) Coherent sources with the EMG signal were detected at fast tapping frequency using a DICS beamformer. (2) Timeseries were extracted from the localized sources using an LCMV beamformer. (3) A sliding window time-frequency analysis was applied to transform these signals into a time-frequency-representation. By averaging over frequencies (14–20 Hz) a low beta band signal was extracted. (4) Source signals were fed into a time and frequency resolved directed connectivity analysis (TPDC).
DOI: https://doi.org/10.7554/eLife.48404.004

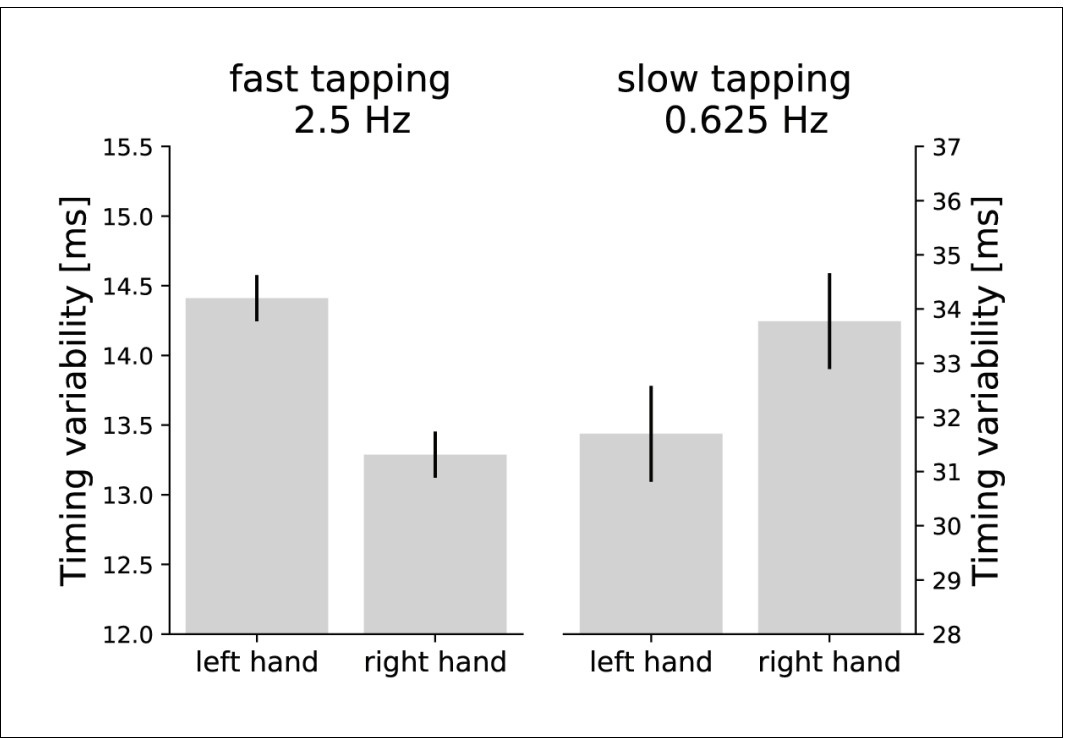

**Figure 3.** Asymmetries in performance. Timing variability is defined as standard deviation of the absolute distance between the actual and target inter-tap-intervals (*Pflug et al., 2017*). Smaller values are associated with better performance. Error bars represent the within subject standard error of the mean. While the right hand taps more precisely in fast tapping, the left hand demonstrates lower timing variability in the slow tapping condition. The interaction between hand and condition is significant at p = 0.003. Note the overall higher precision in fast compared to slow tapping (differently scaled y-axes; *Repp, 2005*).

DOI: https://doi.org/10.7554/eLife.48404.005

The following source data is available for figure 3:

**Source data 1.** Source-data contains for each participant the timing variability for each hand (right and left) and each condition (slow and fast tapping).

DOI: https://doi.org/10.7554/eLife.48404.006

(*Kimura, 1993*; *Haaland et al., 2004*), instead, would predict control of both rhythms by the left hemisphere. Functional specialization of the two hemispheres has not only been linked with lateralized regional activation, but also with the formation of lateralized functional networks of regions (*Stephan et al., 2003*; *Keller and Kell, 2016*). We thus investigated whether auditory-motor interactions between the right and left auditory association cortex and the supplementary motor area (SMA), a motor association area highly involved in the internal generation of sequences (*Kotz et al., 2009*; *Merchant et al., 2013*; *Merchant et al., 2015*; *Crowe et al., 2014*), were modulated differently in the two hemispheres when representing the slow in addition to the fast rhythm. We hypothesized that auditory-motor effective connectivity may differ between the two hemispheres in terms of connection strength in the beta range. Our results identify the left auditory association cortex as the primary cortical area that represents the relative fast auditory rhythm while the right auditory association cortex is recruited to represent the relative slow tapping rate in an amplitude modulation of low beta oscillations. In contrast, motor cortices and the cerebellum only represent the temporal regularities of the motor output. Representing the slow in addition to the fast rhythm increases low beta functional connectivity from the right auditory association cortex to the SMA in parallel to increased BOLD activation of these regions. Further, stronger and bidirectional low beta functional connectivity between the SMA and the left auditory association cortex may privilege the left hemisphere for hierarchical integration of interrelated rhythms in a Gestalt (*Iversen et al., 2008*; *Swinnen and Wenderoth, 2004*).

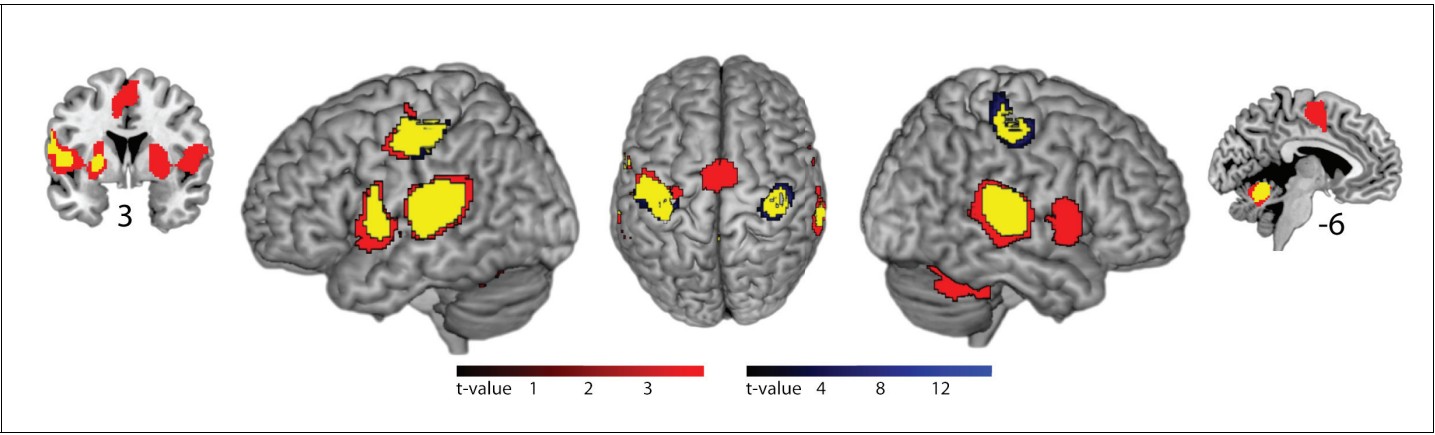

**Figure 4.** Brain areas activated by rhythmic finger tapping. Red: BOLD activation associated with slow tapping (p<0.05, FWE cluster-level corrected). Blue: BOLD activation associated with fast tapping (p<0.05, FWE cluster-level corrected). Yellow: Overlap of activity associated with slow and fast tapping. 3 and −6 indicate coronal and sagittal coordinates, respectively.

DOI: https://doi.org/10.7554/eLife.48404.007

## Results

### Performance measures indicate hemispheric specializations for relative frequencies

Timing variability was defined as standard deviation of the absolute distance between the actual and target inter-tap-intervals (*Pflug et al., 2017*). This measure characterizes internal timing well (*Repp, 2005*). A two-factor repeated measures analysis of variance (ANOVA) on timing variability across condition (slow and fast bimanual tapping) and hand (left and right) revealed expectedly an interaction between condition and hand (F(1,41) = 10.23, p=0.003). Fast tapping was more precise

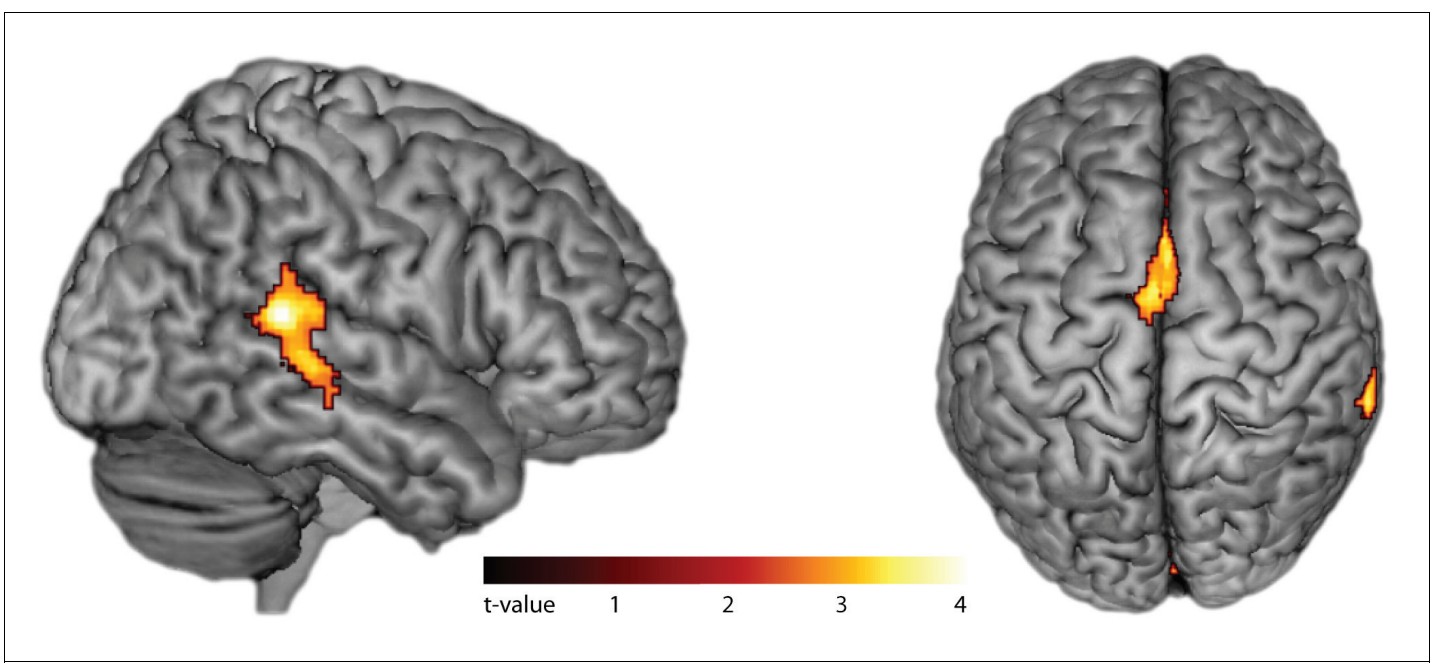

**Figure 5.** Effects of internal generation of a slow rhythm. BOLD activation for slow compared to fast tapping (p<0.05, FWE cluster-level corrected). Activity in the auditory association cortex is right-lateralized at p<0.001.

DOI: https://doi.org/10.7554/eLife.48404.008

**Table 1.** FMRI effects of Slow Rhythm Generation (slow > fast).
BA, Brodmann area; L, left hemisphere; R, right hemisphere.

| Anatomical region | Cluster p-value | Cluster size | Local maxima | BA | Peak MNI-coordinates (x,y,z) |
|---|---|---|---|---|---|
| Frontal | 0.000 | 1087 | R Supplementary Motor Area | 6 | [4 8 52] |
| | | | L Supplementary Motor Area | 6 | [-10 -2 54] |
| | | | L Superior Frontal Gyrus | 6 | [-20 -4 54] |
| | | | R Middle Cingulum | 24 | [2 14 38] |
| | | | R Anterior Cingulum | 24 | [4 28 24] |
| Temporal | 0.001 | 619 | R Superior Temporal Gyrus (A2) | 42 | [62 -38 14] |
| | | | R Middle Temporal Gyrus (A2) | 21 | [46 -38 14] |
| Occipital | 0.000 | 925 | R Calcarine Sulcus | 18 | [4 -88 -8] |
| | | | R Lingual Gyrus | 18 | [6 -74 -4] |
| | | | L Lingual Gyrus | 18 | [ -4 -72 -4] |

DOI: https://doi.org/10.7554/eLife.48404.009

with the right than the left hand (right hand: mean(M)=13.29 ms, standard deviation (SD) = 2.74 ms; left hand: M = 14.41 ms, SD = 2.78 ms) and slow tapping was more precise with the left compared to the right hand (right hand: M = 33.78 ms, SD = 18.42 ms; left hand: M = 31.70 ms, SD = 14.97 ms, see *Figure 3*). No main effect of hand was observed (F(1,41) = 0.760, p=0.388). There was a main effect of condition (F(1,41) = 108.54, p<0.001) with an overall higher precision in fast (M = 13.85 ms, SD = 2.80 ms) compared to slow tapping (M = 32.74 ms, SD = 16.90 ms) (*Repp, 2005*).

## Slow tapping activates the right auditory association cortex

In fMRI, compared to silent baseline, both slow and fast bimanual tapping showed comparable activation patterns of bilateral regions involved in auditory-paced finger tapping, including the primary hand motor cortex, the dorsal and ventral premotor cortex, SMA, the cingulate motor area, parietal operculum, superior temporal cortex including the auditory cortex, posterior superior temporal gyrus and sulcus, the putamen, thalamus, and the superior cerebellum (p<0.05, FWE cluster-level corrected, see *Figure 4*). Activity in the auditory association cortex was right lateralized during slow compared to fast tapping (p<0.001, FWE cluster-level corrected, cluster size 395 voxels) and this was the only cortical patch that showed lateralized activity (all other p>0.05, FWE cluster-level corrected). Generation of the slow rhythm activated additionally the bilateral fronto-mesial cortex including the SMA (see *Figure 5* and *Table 1*).

## Rhythms are differently represented in the left and right auditory association cortex

fMRI revealed a higher activation in the SMA and right auditory association cortex for slow compared to fast tapping. Thus, MEG power spectral densities of both sources, as well as left auditory association cortex, were tested for differences between simple auditory-motor synchronization and additional internal generation of the slow rhythm. In all three areas (SMA and both auditory association cortices), slow compared with fast tapping increased power in the low [14–20 Hz] and high beta band [21–30 Hz] but not in the delta, theta, alpha, or gamma range (see *Table 2*). Condition differences were stronger in the low compared to the high beta band in the SMA and in the right A2 (SMA: t(16) = 3.033, p=0.002, p=0.818, right A2: t(16) = 1.907, p=0.046), but not in the left A2 (left A2: t(16) = 0.228), a region that did not show condition effects in the fMRI. This confirms a more pronounced role of the low compared to the high beta band in rhythm generation (*Gompf et al., 2017*; *Fujioka et al., 2015*). Further analyses were therefore focused on the low beta band. The internal generation of the slow rhythm during slow tapping increased low beta power compared to fast tapping, during which beta power was strongly suppressed, in both auditory association cortices (main

**Table 2.** MEG source power for fast (f) and slow (s) tapping (Mean/Standard Deviation).

| Freq. band | SMA | Left A2 | Right A2 | Left M1 | Right M1 | L cerebellum | R cerebellum |
|---|---|---|---|---|---|---|---|
| theta | f: 0.083/0.107 | f: 0.030/0.046 | f: 0.088/0.167 | f: 0.0673/0.1477 | f: 0.0670/0.1403 | f: 0.050/0.063 | f: 0.057/0.119 |
|  | s: 0.076/0.088 | s: 0.030/0.048 | s: 0.089/0.170 | s: 0.0674/0.1455 | s: 0.0765/0.1777 | s: 0.046/0.056 | s: 0.056/0.105 |
| alpha | f: 0.062/0.077 | f 0.028/0.040 | f: 0.084/0.159 | f: 0.0742/0.1767 | f: 0.0812/0.1977 | f: 0.041/0.045 | f: 0.047/0.080 |
|  | s: 0.073/0.088 | s: 0.027/0.037 | s: 0.090/0.166 | s: 0.0836/0.2052 | s: 0.1025/0.2714 | s: 0.040/0.040 | s: 0.046/0.063 |
| low beta | f: 0.031/0.034 | f: 0.013/0.018 | f: 0.036/0.064 | f: 0.0280/0.0473 | f: 0.0296/0.0533 | f: 0.015/0.014 | f: 0.019/0.025 |
|  | s: 0.039/0.040 | s: 0.014/0.019 | s: 0.040/0.068 | s: 0.0344/0.0615 | s: 0.0390/0.0754 | s: 0.015/0.013 | s: 0.020/0.024 |
| high beta | f: 0.018/0.021 | f: 0.008/0.012 | f: 0.018/0.029 | f: 0.0161/0.0285 | f: 0.0153/0.0277 | f: 0.008/0.008 | f: 0.009/0.011 |
|  | s: 0.022/0.022 | s: 0.008/0.012 | s: 0.020/0.031 | s: 0.0182/0.0318 | s: 0.0189/0.0348 | s: 0.008/0.007 | s: 0.009/0.010 |
| gamma | f: 0.007/0.006 | f: 0.003/0.004 | f: 0.006/0.009 | f: 0.0064/0.0088 | f: 0.0057/0.0088 | f: 0.004/0.004 | f: 0.005/0.005 |
|  | s: 0.007/0.006 | s: 0.003/0.003 | s: 0.006/0.009 | s: 0.0065/0.0090 | s: 0.0059/0.0092 | s: 0.004/0.004 | s: 0.005/0.004 |

DOI: https://doi.org/10.7554/eLife.48404.011

effect of condition F(1,16) = 7.267, p=0.011, permutation ANOVA on mean values over the low beta band). Notably, low beta power condition differences between slow and fast tapping were larger in the right compared to the left auditory cortex (interaction between condition and hemisphere F(1,16) = 3.460, p=0.045) possibly explaining the right-lateralized activation of this cortical region in

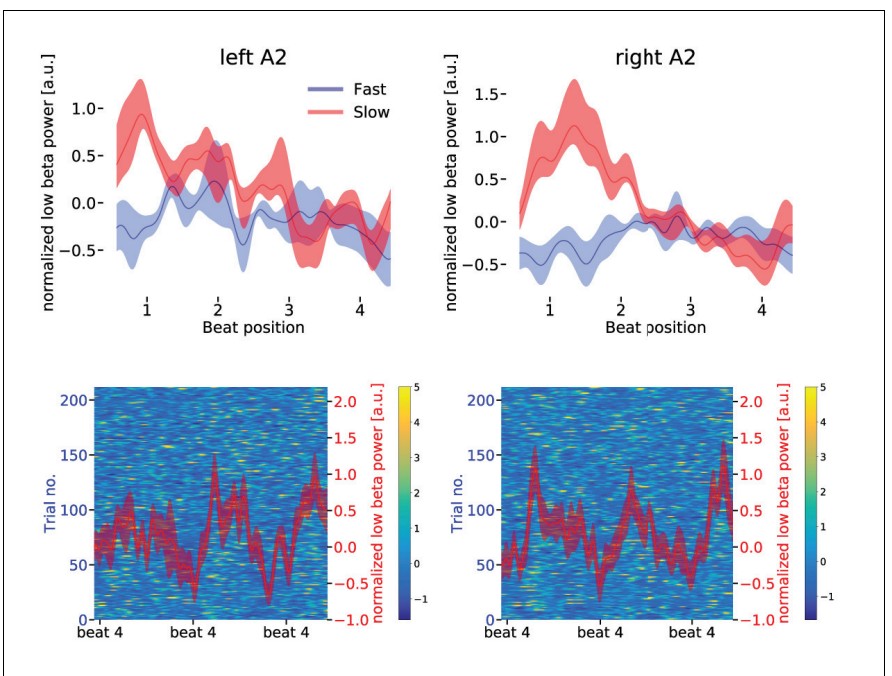

**Figure 6.** Upper panels: Low beta band (14–20 Hz normalized to mean over conditions) power modulation in the left (upper left panel) and right (upper right panel) auditory association cortex (A2) for slow (red) and fast (blue) tapping. One sequence of four auditory beats is illustrated. There was a stronger representation of the fast auditory beat frequency in the left compared to the right auditory association cortex during slow finger tapping (for statistics in the spectral domain, please see main text). Data are aligned to the tap at beat position four. Note the different scales for the beta power in left and right auditory association cortex. Shaded error bars represent the standard error of the mean (SEM). Lower panels: The background illustrates the low beta power in single slow tapping trials. Two sequences of four auditory beats with taps at beat position four are illustrated. Red curves represent mean low beta power ± SEM. Data are aligned to the right beat four in the panels.

DOI: https://doi.org/10.7554/eLife.48404.010

fMRI. During slow tapping, low beta power was maximal at beat position one and decreased to maximal beta suppression at the tap on beat position four in both the left and right auditory association cortex (red curves in *Figure 6*).

While this temporal modulation that reflected the rate of the internally generated slow rhythm was observed in both the left and the right auditory association cortex, the additional temporal modulation at the relative fast auditory beat frequency (2.5 Hz) was stronger in the left than in the right auditory association cortex (red curve in *Figure 6*, upper left panel).

In the spectral domain, this translated to a stronger temporal modulation at the fast auditory beat rate in the left compared to the right auditory cortex during slow tapping (t(16) = 1.8956, p=0.037). In contrast, low beta power modulation at the slow tapping rate was stronger in the right compared to the left auditory association cortex (t(16) = 1.636, p=0.040). In the fast tapping condition, during which participants actively tapped to every auditory beat, beta power was maximally suppressed during the entire sequence of four beats (blue curves in *Figure 6*, upper panels). Consequently, decreases at the single beat positions were less pronounced (*Kilavik et al., 2013*).

## Low beta amplitude modulation in the motor cortices reflects motor output

Power in the low beta band was also less suppressed in the SMA during slow compared to fast tapping (t(16) = 2.0917, p = 0.002; dependent sample permutation t-tests on mean values over the low beta band). In contrast to the auditory cortices, temporal modulation of the low beta power envelope reflected the actual tapping rates (see *Figure 7*). While auditory-motor synchronization in fast

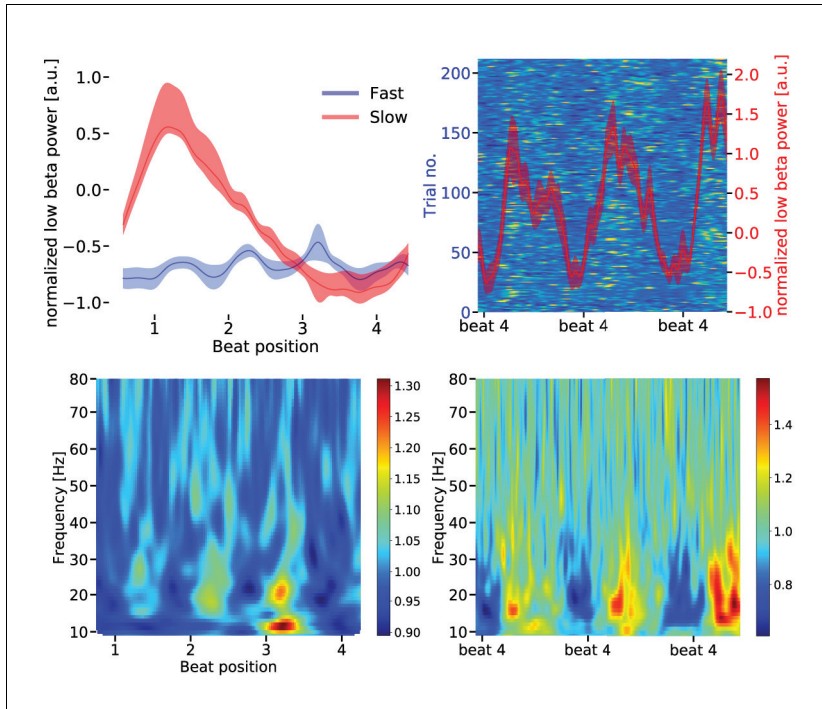

**Figure 7.** Upper left panel: Low beta band [14–20 Hz] power modulation in the supplementary motor area (SMA) for slow and fast tapping (mean over sequences). During fast tapping (blue) the low beta power is modulated by the fast tapping rate while during slow tapping (red) there is a temporal modulation by the slow tapping rate (linear beta power decrease). Data are aligned to tap at beat position four. Shaded error bars indicate the standard error of the mean (SEM). For statistics in the spectral domain, please see main text. Upper right panel: The background illustrates the low beta power in single slow tapping trials. Two sequences of four auditory beats with taps at beat position four are illustrated. Data are aligned to the tap at the right beat position four in the panels. Same scale as in *Figure 6*. Red curves represent mean low beta power ± SEM. Lower panels: Time frequency representation of the SMA source signal during fast (left panel) and slow tapping (right panel).
DOI: https://doi.org/10.7554/eLife.48404.012

tapping decreased low beta power at every beat position, low beta power in the SMA decreased linearly from start of the sequence to the fourth beat position in the slow tapping condition. Consequently, the SMA spectrum contained a strong peak around 0.625 Hz (amplitude = 0.357 a.u.), but no peak at 2.5 Hz during slow tapping. During fast tapping there was a strong modulation at 2.5 Hz (amplitude = 0.695 a.u.) and only a very weak modulation around 0.625 Hz (amplitude = 0.058 a.u.). We further investigated whether the signal in the primary hand motor cortices and the cerebellum resembled the one observed in the SMA. Low beta amplitude modulation at auditory beat frequency during slow tapping did not differ between primary hand motor areas and the SMA (left M1: $t(16)$ = 1.217, $p=0.133$, right M1: $t(16)$ = 0.910, $p=0.182$), between the left and right hand motor cortex ($t(16)$ = 0.899, $p=0.332$) or between the cerebellum and the SMA (left cerebellum $t(16)$ = 1.386, $p=0.095$, right cerebellum $t(16)$ = 1.223, $p=0.110$). Together, in contrast to the auditory association cortices, the primary hand motor cortices, cerebellum and the SMA coded solely the motor output in the amplitude modulation of low beta oscillations.

## Low beta band power modulations explain timing variability

If indeed the low beta power modulation reflects internal timing during slow tapping, it should predict timing variability in single trials. To investigate low beta band differences between short and long inter-tap-intervals during slow tapping, a permutation cluster statistic was used to check for effects of timing variability. Low beta power modulation in the left auditory cortex did not contribute to timing variability during slow tapping (*Figure 8*, leftmost panel). In the right auditory association cortex, the amplitude modulation during too long inter-tap-intervals was larger compared to the power modulation during too short inter-tap-intervals in the sense that low beta power was enhanced at beat position one when participants produced a too long inter-tap-interval (*Figure 8*, left middle panel, significant cluster at 560–660 ms, $p=0.042$). In the SMA, low beta amplitude at beat position one did not influence performance during slow tapping significantly. Instead, too long inter-tap-intervals during slow tapping were associated with a longer low beta suppression at the end of the sequence coinciding with the delayed tap (*Figure 8*, right middle panel, significant cluster at 1400–1470 ms, $p=0.033$; significant cluster at 1560–1800 ms $p=0.001$). During fast tapping, low beta amplitude coded performance in the SMA. A permutation analysis on fast tapping sequences revealed amplitude coding with enhanced beta power modulations for too long inter-tap-intervals and reduced beta power modulations for too short inter-tap-intervals (*Figure 8*, rightmost panel, $p=0.002$). Due to the maximal low beta suppression in auditory cortices during fast tapping, no significant difference between too long and too short inter-tap-intervals was observed in these regions.

## Auditory-motor interactions

To study the contribution of auditory-motor interactions to the right-lateralized processing of the slow rhythm during slow compared to fast tapping, time-resolved partial-directed coherence (TPDC)

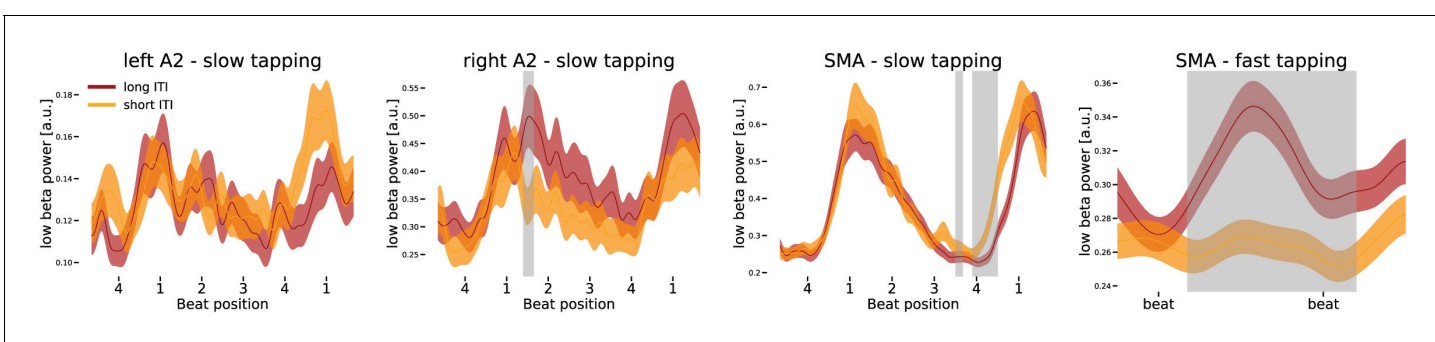

**Figure 8.** Differences in low beta power modulation between too short and too long inter-tap-intervals (ITI). Sequences of four auditory beats with taps at beat four were aligned at the left tap in the first three panels. While the first three panels illustrate effects during slow tapping, the right panel illustrates low beta power in the supplementary motor area (SMA) during fast tapping (data left aligned). Significant differences between too long and too short sequences were marked in grey. Only low beta power in the right auditory association cortex (A2) and in the SMA predicted performance during slow tapping. Low beta amplitude coded the ITI during fast tapping in the SMA. Note the different scales in the panels.
DOI: https://doi.org/10.7554/eLife.48404.013

was calculated between the secondary auditory cortices and the SMA and vice versa on MEG source level data. This measure is insensitive to local power differences (*Kaminski et al., 2016*; *Tsapeli and Musolesi, 2015*; *Nalatore et al., 2007*; *Muthuraman et al., 2018*) and is ideally suited to investigate time-resolved directed functional connectivity. Both slow and fast tapping increased TPDC between the auditory cortices and the SMA in the low beta and mid gamma range with strongest effective connectivity from the left auditory association cortex to the SMA (see *Figure 9*).

To reveal directed connectivity when representing two rhythms instead of one rhythm, we focused the connectivity analyses on the contrast between slow and fast tapping and restricted them again to the low beta band (*Gompf et al., 2017*; *Fujioka et al., 2015*).

A two-factor repeated measures ANOVA on averaged connectivity in the low-beta band across hemisphere (left and right) and direction (auditory to motor and motor to auditory) revealed a main effect of hemisphere (F(1,16) = 7.00, p=0.018) with stronger condition differences between slow and fast tapping in the left (M = 0.007, SD = 0.0048) compared to the right (M = 0.003, SD = 0.0054) hemisphere. This surprising effect was accompanied by a close-to-threshold interaction between direction and hemisphere (F(1,16) = 3.83, p=0.068). While the connections from left A2 to the SMA (t(16) = 4.174, p=0.002), from the right A2 to the SMA (t(16) = 2.988, p=0.005), and the one from the SMA to the left A2 (t(16) = 3.385, p=0.001) increased low-beta connectivity for slow compared to fast tapping, the connection from the SMA to the right A2 was not enhanced for slow compared to fast tapping (t(16) = −0.882, p=0.392, see *Figure 10*, left panel).

In sum, slow compared with fast tapping increased interactions in the low beta band between both the left and right auditory association cortex and the SMA and between the SMA and the left auditory association cortex (see *Figure 10*, right panel) with an overall stronger connectivity in the left compared to the right hemisphere. We investigated individual timing variability in the slow tapping condition for a correlation with directed connectivity contrasts for slow > fast tapping. An increased connection strength in the connection from the SMA to the left A2 during slow compared to fast tapping correlated twith timing variability in the sense that it reduced timing variability of the right hand when tapping slowly (r = −0.490. p=0.04). All other correlations were not significant (p>0.05).

Auditory-motor interactions were not only structured in frequency, but also in time (see *Figure 9*). The effective connectivity in the low beta range for slow compared to fast tapping was amplitude-modulated by a theta rhythm at 6.5 Hz in all connections except for the connection from the SMA to the right auditory association cortex (for statistics see *Table 3*), the connection that also did not show significant low beta band condition effects. There was an additional modulation of effective connectivity in the low beta range by an alpha rhythm at 10.5 Hz in all connections except for the connection from the right auditory cortex to the SMA. There was no other rhythmic modulation of low beta effective connectivity for slow compared to fast tapping (all p>0.05).

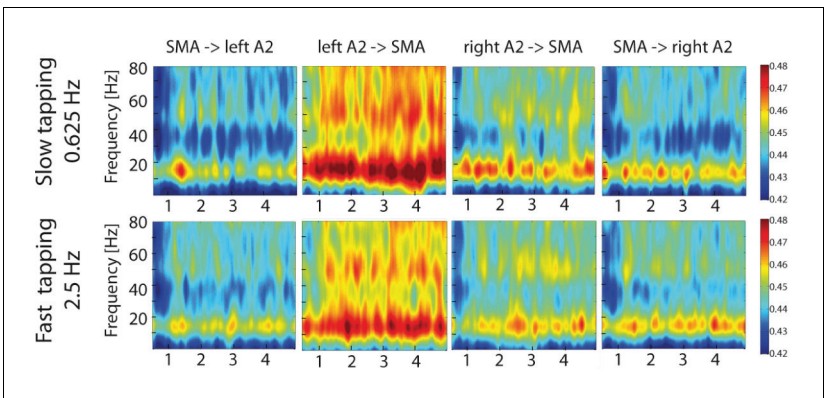

**Figure 9.** Time-resolved partial-directed coherence (TPDC) during slow (upper panels) and fast tapping (lower panels). TPDC was particularly strong in the low beta and the low gamma band. Note the overall increased connectivity strength between the left auditory association cortex and the SMA compared to the other connections. SMA: supplementary motor area. A2: Auditory association cortex.
DOI: https://doi.org/10.7554/eLife.48404.014

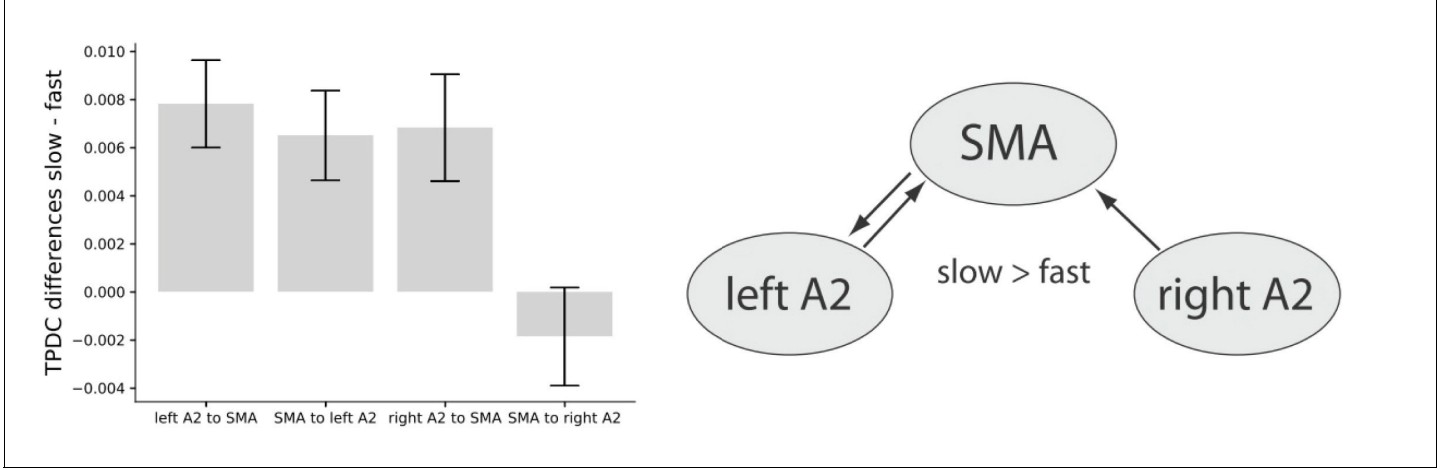

**Figure 10.** Left panel: Condition differences between slow and fast tapping in low beta band [14–20 Hz] effective connectivity.
Right panel: Connections with increased low beta band interactions (p<0.05) during slow compared to fast tapping. SMA: supplementary motor area.
A2: Auditory association cortex. TPDC: Time-resolved partia-directed coherence.
DOI: https://doi.org/10.7554/eLife.48404.015

The following source data is available for figure 10:

**Source data 1.** Source-data contains for each participant the mean low beta connectivity from the TPDC for the four connections (right A2 to SMA, SMA to right A2, left A2 to SMA and SMA to left A2).
DOI: https://doi.org/10.7554/eLife.48404.016

## Discussion

Our experiments identify the auditory association cortex as the part of the brain that represents relative rhythms differently in the two hemispheres. During auditory-paced slow bimanual tapping, when two interrelated rhythms have to be represented by the brain, the left auditory association cortex represents the fast auditory beat rate more strongly than the right auditory association cortex in an amplitude modulation of low beta oscillations, although both auditory cortices receive identical sensory input. While both auditory association cortices represent also the internally generated slow tapping rate in an amplitude modulation of low beta power, the right auditory association cortex increases low beta power more strongly for slow compared to fast tapping and activates more strongly in terms of BOLD than the left auditory association cortex. This suggests that the brain represents the faster rhythm preferentially in the left and the slower rhythm preferentially in the right hemisphere which ultimately results in different tapping precision of the left and right hand during slow and fast tapping, respectively.

### Rhythm representations in the auditory association cortices

A functional lateralization in terms of differences in activation of functional homologues was only observed in the auditory and not in the motor association cortices. Together with the cerebellum, motor cortices rather mirrored the actual motor output with a stronger BOLD signal in the bilateral SMA associated with reduced beta suppression during internal timing compared to auditory-motor

**Table 3.** Rhythmic modulation of the low beta band directed connectivity in slow compared to fast tapping.

| Connection | Modulation peak in theta band | Modulation peak in alpha band |
| --- | --- | --- |
| left A2 to SMA | t(16)=2.441, p<0.001 | t(16)=1.803, p=0.031 |
| right A2 to SMA | t(16)=2.305, p=0.001 | t(16)=1.399, p=0.114 |
| SMA to left A2 | t(16)=2.760, p=0.004 | t(16)=2.398, p=0.002 |
| SMA to right A2 | t(16)=1.509, p=0.083 | t(16)=2.093, p=0.010 |

DOI: https://doi.org/10.7554/eLife.48404.017

synchronization (*Gompf et al., 2017*). Beta suppression during slow tapping was not maximal such that ceiling levels cannot explain missing lateralization in the SMA. Lateralization in motor association cortices is often observed in the lateral dorsal premotor cortex, particularly during asymmetric or complex bimanual actions as compared to the symmetric finger taps used in this study (*Haslinger et al., 2002*; *Hlustík et al., 2002*; *Hardwick et al., 2013*). The lateral dorsal premotor cortex is activated by polyrhythmic external stimuli while internally generated rhythms activate the SMA (*Swinnen and Wenderoth, 2004*). The lack of lateralization effects in the motor cortices in our study suggests that the observed functional lateralization on the behavioral level was timing-related and not related to bimanual motor coordination (*Serrien et al., 2003*).

Indeed, right lateralization of auditory association cortex activity was related to improved left hand timing despite a bilateral activation of the SMA in slow tapping. No other brain areas beyond the bilateral SMA and the right auditory association cortex activated significantly for slow compared with fast tapping, a condition that could have been associated with increased counting effort compared to the fast tapping condition. Counting during perceptual grouping activates the intraparietal sulcus, dorsolateral prefrontal and inferior frontal cortex (*Ansari, 2008*), none of which was activated in our study. Our results confirm a strong contribution of the sensory cortices to the lateralization of action control, as suggested by the sensory-driven hypotheses on hemispheric specialization (*Minagawa-Kawai et al., 2011*; *Ivry and Robertson, 1998*). Because both auditory cortices receive the same auditory input (the fast auditory beat rate) in both slow and fast tapping, the nearly absent fast auditory beat rate representation in the right auditory association cortex likely constitutes the consequence of dynamic attention to every fourth auditory stimulus. Notably, this filtering is performed in the temporal domain, suggesting that the right auditory association cortex actively selects the relevant auditory beats for slow rhythm generation. This is reminiscent of the dynamic attending theory (*Jones, 1987*) which proposes that during perception, auditory cortex oscillations are aligned to rhythmic auditory input to select behaviorally relevant input (*Schroeder and Lakatos, 2009*).

Beta power in the right but not in the left auditory association cortex explained timing variability during slow tapping. The internal generation of a too slow rhythm was associated with an even larger amplitude modulation with enhanced beta power at beat position one. The same amplitude coding was observed in the SMA, although only during fast tapping. In our experiment, time information was coded in the amplitude of beta oscillations. Recently, time information during rhythmic finger tapping in the subsecond range has also been related to the amplitude of abstract representations of the SMA neural population dynamics in non-human primates (*Gámez et al., 2019*). Neither did the neuronal population dynamics scale in time, nor was the slope of the beta power modulation in our study modulated by slow vs. fast finger tapping, which suggests time indeed is coded in amplitude, at least during rhythmic finger tapping (see *Fujioka et al., 2015* and *Wang et al., 2018* for contrasting views). Assuming time information is coded accumulator-like (*Ivry and Richardson, 2002*) in the power difference between minimal and maximal beta suppression we may state that amplitude coding identifies brain regions with different preferred time intervals. The association between timing variability and amplitude coding in the motor association cortex during fast tapping and the relationship between timing variability and amplitude coding in the right but not left auditory association cortex during slow tapping confirms that the brain uses the motor system for subsecond timing and non-motor cortices for suprasecond timing (*Morillon et al., 2009*). During slow tapping the SMA coded solely the information on the actual timing of the tap in the latency of the maximal suppression at beat position four. This emphasizes the contribution of the right auditory association cortex to the internal generation of a slow, supra-second rhythm.

Low beta power in the cerebellar sources did only mirror the motor output in our experiment. However, subcortical regions including the cerebellum, basal ganglia and thalamus, make part of a dedicated neural timing system and likely provide more than motor timing information (*Kotz and Schwartze, 2011*). We cannot rule out that other cerebellar sources with less cortico-muscular coherence compared to the cerebellar sources identified here show internal timing-related profiles.

## Polyrhythmicity during syncopation activates the right auditory association cortex

Although the right auditory association cortex was more strongly activated by the slow than the fast tapping condition, also the left auditory association cortex represented the slow tapping rate in a beta power decrease from start to the end of a sequence of four auditory beats. This raises the

question why the right auditory association cortex was additionally recruited for slow tapping and associated with performance if all necessary temporal information could be decoded from left auditory association cortex. Note that in our study participants tapped the slow rhythm on every fourth auditory beat, which represents a syncopated rhythm with a 270° phase delay in relation to the 4/4 standard meter that was introduced by the four priming auditory beats prior to each trial. In a previous behavioral experiment, we showed that the right hemisphere advantage for the control of slow tapping depended on syncopation, because it was not observed when participants tapped at auditory beat position one when tapping slowly (at 0° phase difference relative to the standard meter; *Pflug et al., 2017*). Non-syncopated slow tapping at beat position one reflects the overlearned 4/4 meter that constitutes the standard meter in Western culture (*London, 2012*). In dynamic pattern theory, 0° phase angles represent more stable dynamical states compared to 270° phase differences (*Zanone and Kelso, 1992*).

Perceiving syncopated compared to non-syncopated rhythms activates the right more than the left auditory association cortex (*Herdener et al., 2014*). However, our results are not a consequence of stimulus features like accentuation or lengthening, because the auditory stream used in this experiment consisted of identical beats. They rather indicate that the involvement of the right auditory cortex is not a direct consequence of the increased complexity of rhythmic grouping during syncopated slow tapping compared to simple auditory-motor synchronization during fast tapping. Instead, the MEG and behavioral results demonstrate that the two rhythms are not represented randomly in the left and right hemispheres, but rather systematically with a stronger representation of the relative fast rhythm in the left and of the relative slow rhythm in the right hemisphere.

We interpret our observation in such a way that syncopated rhythms are represented separately by the two hemispheres as long as they are not yet hierarchically integrated in a Gestalt based on experience. From a dynamic pattern theory perspective, the 270° phase angle during syncopated tapping induces competition between attractor states with a strong tendency to tap at 0° phase difference relative to the standard meter (*Swinnen, 2002*). This tendency could be reduced by increasing the energy needed for a phase transition from tapping the instructed 270° phase angle to the tapping along the standard meter. Note that in motor as well as auditory cortices, beta amplitude was minimal at beat position four and maximal at beat position one during slow tapping, which decreases tapping probability at beat position one. Competition between the standard meter and the phase-shifted slow tapping rhythm of same frequency could be reduced by increasing the physical distance of their representations. This would permit parallel representations of competing attractors. The brain could potentially solve this problem by representing the standard meter and its relationship to the fast auditory beat rate in the right hemisphere and the slow tapping rhythm in the left hemisphere. The fact that the brain does not follow this path and rather represents the internally generated phase-shifted slow rhythm in the right auditory association cortex and the fast auditory beat rate in the left auditory association cortex speaks in favor of different temporal filters in auditory association cortex as sources of hemispheric specialization.

In addition, preferential binding of rhythms with different frequencies in the left hemisphere may contribute to functional lateralization. The left hemisphere outperforms the right hemisphere in local binding (*Flevaris et al., 2010*), which is critical for beat and meter integration. Integrating fast rhythms and slow rhythms with 0° phase angle relative to the standard meter may facilitate hierarchical binding in a Gestalt (*Zanone and Kelso, 1992*) which could bias meter processing to the left hemisphere. This may explain the numerous reports on an involvement of the left hemisphere in rhythm production in professional musicians (*Vuust et al., 2006*; *Kunert et al., 2015*; *Herdener et al., 2014*) and explain empirical findings that ostensibly support the motor-driven hypotheses of left hemispheric dominance (*Kimura, 1993*; *Haaland et al., 2004*).

## Auditory-motor interactions in the beta range privilege the left hemisphere for rhythm integration

Tapping rhythms were most strongly represented in the low beta band in both motor and auditory cortices. Effects in the beta band have often been found in tasks that require synchronization of large-scale brain networks (*Gehrig et al., 2012*; *Roelfsema et al., 1997*) and more recently have been associated with top-down signals in hierarchically organized cortical networks (*Bastos et al., 2015*; *Fontolan et al., 2014*). Beta oscillations are particularly strong in the motor system including the basal ganglia, which also play an important role in rhythmic motor behavior (*Kotz et al., 2009*).

During finger tapping, spike-field coherence in the striatum is stronger for beta compared to gamma oscillations and beta oscillations are more strongly related to internal rhythm generation than with sensory processing during tapping (*Bartolo et al., 2014*). Consequently, an important role of auditory-motor interactions in the beta range was expected and, indeed, interactions between the auditory and motor association cortices were strongest in the low beta band. However, the internal representation of the slow tapping rhythm in the right auditory association cortex was not associated with additional top-down signals in the low beta band from the SMA to the right auditory association cortex compared to fast tapping during which no additional rhythm was represented.

In the right hemisphere, slow tapping increased information flow in the low beta band only from the auditory association cortex to the SMA, possibly to provide slow rhythm information. The SMA received also stronger low beta input from the left auditory association cortex during slow compared to fast tapping which could reflect the effort to integrate the slow with the fast rhythm that was more strongly represented in the left auditory association cortex. The SMA could thus be interpreted as the midline structure that integrates rhythm information from both auditory association cortices and times tapping accordingly. Yet, in contrast to the right hemisphere, slow tapping also strengthened the top-down connection from the SMA to the left auditory association cortex compared to fast tapping. This left-lateralized top-down connection was the only connectivity in our study that correlated with tapping accuracy. The stronger the connection was from the SMA to the left auditory association cortex during slow compared with fast tapping, the more precise participants tapped with their right hand in the slow tapping condition. This suggests that the right hand, that taps fast rhythms more precisely than slow rhythms, may benefit from bidirectional auditory-motor interactions in the left hemisphere when tapping slowly. Together with the overall stronger directed connectivity between the left auditory association cortex and the SMA, this auditory-motor loop may facilitate rhythm integration in the left hemisphere (*Nozaradan et al., 2015*). Auditory-motor loops can also be used to facilitate perceptual timing in the absence of overt motor behavior. Such a motor facilitation is efficient when estimating time periods of below one to two seconds (*Morillon et al., 2009*; *Rao et al., 1997*; *Funk and Epstein, 2004*). Auditory rhythmic sampling without overt motor behavior involves beta signals from the left lateralized motor cortex to the auditory association cortex (*Morillon and Baillet, 2017*). This finding confirms the left-lateralized top-down connection in beta connectivity between the motor and auditory association cortex found in our study even in the absence of overt movement.

Beta signals associated with slow compared to fast tapping between the auditory association cortices and the SMA and vice versa were modulated by a theta rhythm. Fronto-temporal theta oscillations have been associated with auditory-motor and multisensory integration (*van Atteveldt et al., 2014*) and more specifically support auditory working memory (*Albouy et al., 2017*), speech perception (*Assaneo and Poeppel, 2018*), and speech production (*Behroozmand et al., 2015*). The auditory-motor theta rhythm observed in this finger tapping study was observed at a peak frequency of 6.5 Hz, which corresponds to the frequency at which also the velocity of slow finger movements is modulated (*Gross et al., 2002*). This is slightly higher than the auditory-motor theta rhythm associated with speech processing, which peaks at 4.5 Hz (*Assaneo and Poeppel, 2018*), potentially due to the higher natural resonance frequencies of the fingers compared to the jaw (*Junge et al., 1998*).

## Implications for speech processing

The observed functional differences between the hemispheres during auditory-paced finger tapping remind the asymmetries observed during speech processing. During speech perception, the syllable rate in the theta range serves as a strong acoustic cue that entrains oscillations in the bilateral auditory association cortex and induces auditory-motor interactions in this frequency range (*Assaneo and Poeppel, 2018*). Auditory-motor interactions are left-lateralized both during speech perception (*Mottonen et al., 2014*; *Murakami et al., 2015*; *Hickok, 2015*) and speech production (*Kell et al., 2011*; *Keller and Kell, 2016*) suggesting left-lateralized auditory-motor loops. In both motor and auditory association cortices, binding of speech-relevant rhythms via cross-frequency coupling is left lateralized in fronto-temporal cortices during speech perception (*Gross et al., 2013*). These observations suggest left-lateralized auditory-motor loops could improve rhythm integration by cross-frequency coupling both during speech perception and production. Indeed, reduced auditory-motor coupling in the left hemisphere of people who stutter is associated with overt deficits in controlling speech rhythm (*Neef et al., 2015*; *Chang and Zhu, 2013*; *Kell et al., 2018*). The deficit

in rhythm integration in people who stutter is associated with an over-recruitment of the right hemisphere during speech production that is reduced upon recovery (*Kell et al., 2009*; *Kell et al., 2018*). The right over-activation during speaking may be regarded as a strategy to separate competing attractors that arise from insufficient auditory-motor mapping in the left hemisphere (*Hickok et al., 2011*). It is interesting to note in this context that this speech disorder that has been associated with basal ganglia dysfunction (*Alm, 2004*) shows abnormal beta oscillations associated with timing both in speech and non-speech tasks (*Etchell et al., 2016*; *Etchell et al., 2014*). Future research will need to elucidate the commonalities and differences between the functional lateralization of hand control and the lateralization of speech production.

## Conclusions

We show here that representing two rhythms during syncopation lateralizes processing of the relative faster rhythm to the left and the relative slower rhythm to the right hemisphere. Auditory association cortices filter adaptively the preferred temporal modulation rate and send this time signal to the SMA for motor output coordination. The filter is relative rather than absolute, meaning that the hemispheres do not lose the complementary information, but nevertheless represent preferentially different rhythms. An additional top-down communication from the SMA to the left auditory association cortex may privilege the left hemisphere in integrating multiple rhythms in a multiplexed Gestalt, which likely represents a prerequisite for more complex cognitive functions like speech processing.

# Materials and methods

## Participants

Twenty-five participants (10 males; aged 19–31 years; M = 24 years) were included in the fMRI study; seventeen participants (six males, aged 21–38 years; M = 26 years) in the MEG study. Number of participants was chosen based on a literature research for finger tapping experiments in MEG/EEG studies and fMRI, respectively. Participants had normal or corrected-to-normal visual acuity, normal hearing, no neurological deficits and were right-handed according to self-reports and their laterality index based on the Edinburg inventory of manual preference (fMRI: M = 86; MEG: M = 89; *Oldfield, 1971*). Participants performed a test run of approximately five minutes before measurement to become familiar with the task. All participants gave written informed consent prior to the study and were paid for participation. Experimental procedures were approved by the ethics committee of the medical faculty of Goethe university (GZ 12/14), and are in accordance with the declaration of Helsinki.

## Tapping paradigm

In every trial of this auditory paced finger tapping paradigm, 36 auditory beats (1600 Hz, 2 ms) were presented with a constant inter-beat-interval of 400 ms (2.5 Hz, 210 bpm). Participants were asked to tap with their index fingers at two different rates to these beats. No auditory feedback was provided. Thus, auditory input did not differ between conditions. In the fast tapping condition, participants tapped to every auditory beat. In the slow tapping condition, participants were instructed to iteratively count four beats internally and tap only on beat position four (*Figure 1*). Therefore, tapping to the 36 auditory beats resulted in 36 taps when tapping the fast rate and nine taps when tapping the slow rate during each 15 second-long trial. Eight different tapping conditions were performed. In four unimanual conditions participants tapped with one hand (left or right) either the fast or the slow tapping rate. Four bimanual conditions were either performed monofrequent, in which both hands tapped the same rate or multifrequent, in which one hand tapped the fast rate and the other hand the slow rate. Here, we report the two bimanual monofrequent conditions during which both hands were engaged in the same motor output. While 'fast tapping' represents simple auditory motor synchronization to the presented auditory beat, in 'slow tapping' the single beats had to be cognitively grouped in sequences of four. This necessitates the generation of a slower rhythmic structure in addition to processing the same fast auditory beats as in fast tapping. The task was performed in runs with 24 trials each, in semi-randomized order. A trial started with a presentation of a visual instruction that indicated the upcoming condition. Before participants started

tapping, four auditory beats of higher pitch primed the auditory beat rate. The inter-trial interval was jittered in both recordings (fMRI: 10.1–13.6 s and MEG: 7.7–12 s) to reduce temporal predictability during baseline.

### Timing variability

Timing variability was calculated using the standard deviation of absolute distance between the actual inter-tap-intervals and the target inter-tap-interval (400 ms for fast, 1600 ms for slow tapping rates) of consecutive taps (for a more detailed description see *Pflug et al., 2017*). For tap detection, the maximal tap pressure was used. Timing variability was calculated for every hand and condition independently. Values of both recording methods (fMRI and MEG) were used as dependent variables in a 2 (condition [slow, fast]) x 2 (hands [left, right]) mixed design repeated measure analysis. Significant effects were post-hoc tested using paired-sample t-tests. Significance level (alpha) was set at 0.05. Statistical tests were conducted using SPSS Statistics 22.0 (IBM Company, RRID:SCR_002865).

### fMRI

#### Recording procedure

Participants laid in a supine position. Two pneumatic pressure sensors (MP150, Biopac Systems, RRID:SCR_014279) were attached to the pads of participant's index fingers and participants were asked to tap with the sensors on their ipsilateral thigh. The pressure sensitivity of the sensors was 0.01 cm H2O with a sampling rate of 1 kHz. Auditory beats were presented binaurally via headphones and visual condition cues were displayed with a projector on a screen in front of the participants. Pacing signals and visual instructions were presented with Presentation Software (Neurobehavioral Systems, Albany, CA, USA Albany). The fMRI experiment consisted of 2 runs, each including three trials of every condition in randomized order with a one minute break in between runs.

#### Data acquisition

The fMRI data were recorded in a simultaneous EEG/fMRI experiment; however, we only report the fMRI results in the present study. The results of the EEG analysis are presented in *Gompf et al. (2017)*. The entire equipment was fMRI compatible and met all security standards (Brain Products EEG/fMRI Hardware, RRID: SCR_009443). Scanning was performed on a Siemens Trio 3 Tesla magnetic resonance system (Siemens MAGNETOM Vision, Erlangen, Germany) equipped with a circular polarized Send/Receive head coil with an integrated preamplifier. Functional images were obtained with a gradient-echo T2*-weighted transverse echo-planar imaging (EPI) sequence (614 volumes; repetition time (TR) = 2.08 s; echo time (TE) = 29 ms; flip angle = 90°; 32 axial slices; 3mm3 isotropic voxel size). Additionally, high-resolution T1-weighted anatomical scans (TR = 2.25 s; TE = 3.83 ms; flip angle = 9°; 176 slices per slab; 1mm3 isotropic size) were obtained.

#### Preprocessing

Image processing and data analyses were performed in SPM12 (Welcome Trust Centre for Neuroimaging, London, UK; RRID:SCR_007037). After eliminating the first four volumes in each participant due to field inhomogeneity in the beginning of each run, standard preprocessing was performed (realignment, co-registration of anatomical T1-images to the mean functional image with subsequent segmentation using Tissue Probability Maps, normalization to the Montreal Neurological Institute [MNI] standard brain template and smoothing with an 8 mm full-width at half-maximum Gaussian kernel). For lateralization analyses (see below), images were preprocessed with the use of a symmetrical template to prevent anatomical differences between hemispheres from inducing spurious lateralization effects (*Keller and Kell, 2016*). To this end, symmetric Tissue Probability Maps were created by averaging the mean Tissue Probability Map and its flipped counterpart. The preprocessed images were analyzed within the framework of general linear models for time-series data (*Worsley and Friston, 1995*).

## Statistical analysis

On the single-subject level, eight condition-specific regressors were modelled by convolving the onsets and durations of conditions (modelled by boxcar functions) with the canonical hemodynamic response function to obtain predicted BOLD responses. Additional fifteen regressors of no interest were calculated, eight of them capturing the variance associated with the condition instructions, one for modelling an additional tap participants usually made after the last metronome click in fast conditions, as well as six regressors for head-motion-related effects. For group-level analyses, the regressors of all eight conditions, modelling condition-specific tapping effects, were included in a 1 × 8 ANOVA to account for the variance caused by the other conditions. We report the effects of slow and fast bimanual tapping and thus contrasted the conditions separately against an implicit baseline and against each other. The significance threshold was set at p<0.05 family-wise error (FWE) corrected at cluster level with a cluster identification threshold of p<0.001, uncorrected.

## Lateralization analysis

Post-hoc lateralization analyses were performed on the effect of slow rhythm generation. The processing steps followed *Keller and Kell (2016)*. First, contrast images of slow > fast tapping were calculated on the single subject level using the symmetrical template for normalization. These images were then flipped over their mid-sagittal axis. In a subsequent two-sample t-test, the non-flipped contrast images were compared with their flipped counterparts on the group-level. To avoid artifacts resulting from cerebrospinal fluid in the sagittal sulcus, a midline mask was applied. The threshold for significance was set at p<0.05 FWE corrected at cluster level with a cluster identification threshold of p<0.001. This identified significant activity differences between brain regions and their homotopes.

## MEG

### Recording procedure

Participants sat in an upright position in the MEG chamber. Head movements were limited using foam pads. Participants tapped on pressure sensors fixed on the left and right armrests. Auditory beats were presented binaurally through plastic tubes and a projector was used to display visual condition cues on a screen. Participants were asked to restrict their gaze to the center of the screen during the task. The experiment consisted of four runs, 11 min each, including each three trials of every condition, with two minutes breaks in runs. Surface electromyogram (EMG) electrodes were placed over both extensor digitorum communis muscles. Electrooculogram (EOG) was recorded to detect horizontal and vertical eye-movements and electrocardiogram (ECG) for heart beats. Participants' head positions relative to the gradiometer array were determined continuously using three localization coils (ear-channel and nasion).

### Data acquisition

MEG was recorded using a whole-head system (Omega 2005; VSM MedTech) with 275 channels at a sampling rate of 1200 Hz in a synthetic third-order gradiometer configuration. Data were filtered off-line with fourth-order Butterworth 300 Hz low-pass and 2 Hz high-pass filters. Line noise at 50 Hz was removed using a band pass filter. Recorded data were down-sampled to 1000 Hz and fused with the data of the tapping pressure sensors.

### Preprocessing

MEG recordings were preprocessed and analyzed using the Fieldtrip toolbox (*Oostenveld et al., 2011*; RRID:SCR_004849) in MATLAB (RRID:SCR_001622). Trials containing muscle or SQUID artifacts were removed using an automatic artifact rejection algorithm (*Delorme et al., 2004*). Trial segments with a head movement exceeding 5 mm were also discarded from further analysis. Independent component analysis was performed to identify and reject components of heart muscle and blinks. For easier detection, components were correlated with the ECG and EOG signal. Data from whole trials were used for source reconstruction (see next section). Clean trials without artifacts (overall 89%) were cut into sequences containing four consecutive beats. In the fast tapping condition, four consecutive taps were defined as a valid sequence. Sequences in the slow tapping condition were valid if they included only one tap at the intended beat position four. To visualize the slow

rhythm over multiple sequences, we also cut trials into time windows containing two sequences of four auditory beats. To this end we used the taps of the fourth and fifth sequence within each trial, which are in the middle of in total nine sequences per trial. An overview of the following analysis steps is illustrated in *Figure 2* and explained in detail below.

## Source reconstruction

We identified brain areas that showed significant coherence with the averaged EMG signals of the extensor digitorum muscles during bimanual fast tapping following the standard procedure for source analysis during auditory paced finger tapping (see *Figure 2* step 1, (*Gross et al., 2001*; *Pollok et al., 2005*; *Muthuraman et al., 2014*; *Muthuraman et al., 2018*). A spherical MEG source model was used to estimate the sources in every participant. Afterwards, cortico-muscular coherence was calculated and significant sources were identified for every participant individually. In detail, a dynamic imaging of coherent sources (DICS) beamformer approach was used to identify (sub-)cortico-muscular coherent sources. For beamforming, the individual head models were co-registered with the Talairach standard MRI brain using Fieldtrip. MEG source analysis with a spherical head model and further source reconstruction based on a template brain is accurate (*Steinstraeter et al., 2009*) and constitutes the standard approach when individual structural MRI data are not available for every participant (*Jensen et al., 2005*; *Ross and Tremblay, 2009*; *Fujioka et al., 2010*; *Fujioka et al., 2015*), because the spatial precision of detected sources is equivalent to analyses based on individual structural MRI data (*Hasnain et al., 1998*). Since significant coherent sources were identified in each individual, the topographical relationship between sources was additionally considered in source denomination.

Original coherence was tested against 999 surrogate datasets where the MEG trial structure was shuffled but the EMG trials were kept stable. For every grid point, a threshold for detecting coherent sources was set at the 95th percentile of permutation results. This analysis revealed cortical and subcortical brain areas involved in bimanual finger tapping in each participant, including the SMA and thalamus, as well as the bilateral dorsolateral prefrontal cortex, dorsal premotor cortex, primary hand sensorimotor cortex, secondary somatosensory cortex in the Rolandic operculum, posterior parietal cortex, secondary auditory cortex in the superior temporal gyrus, and cerebellum. Since the activation of the right hand motor cortex in bimanual fast tapping was sub-threshold in some participants (above 90th percentile), the unimanual left hand fast tapping condition was used to confirm the localization of the right hand motor cortex above significance level.

## Source level analysis

As a next step, time courses of EMG-coherent sources were extracted separately for the slow and fast bimanual tapping condition using a linearly constrained minimum variance beamformer method (LCMV; *Van Veen et al., 1997*) for each frequency from 2 to 300 Hz (temporal resolution: 1000 Hz, see *Figure 2* step 2). Source level analyses were performed on extracted timeseries of the SMA and the right auditory association cortex, since these areas were identified by fMRI as being activated in slow compared to fast tapping. To compare activity in the right auditory cortex with activity in its homologue, activity in the left auditory association cortex was also extracted. For completeness, also activity in the left and right primary handmotor cortex, as well as activity in both cerebellar hemispheres was investigated.

All signals were converted into the time-frequency space. Wavelet transformations (slepian windows) were calculated for every sequence by sliding a window of 7-times the related period length (2–80 Hz, 1 Hz resolution) in 10 ms steps. Averages over sequences of four auditory beats were calculated for every participant and condition, resulting in two dimensional arrays with time and frequency dimensions (see *Figure 7*, lower panels for examples). Power averages over frequencies were calculated in predefined standard frequency bands: theta [4–7 Hz], alpha [8–13 Hz], low beta [14–20 Hz], high beta [21–30 Hz] and gamma [31–80 Hz]. To compare power spectral differences (PSD) between conditions within cortical sources, frequency bands were averaged over the time domain resulting in one PSD value per condition and participant. These values were used in non-parametric permutation dependent t-tests (999 permutations, alpha = 0.05) to detect condition differences in the amplitude of the predefined frequency bands. Additionally, t-contrasts between conditions from the sources of interest (SMA and auditory cortices) were calculated to test for power

differences between the low and high beta band. Since condition contrasts were higher in the low compared to the high beta band, we focused further analysis on the low beta band (*Figure 2*, step 3), which has been shown to reflect internal timing best (*Fujioka et al., 2015*; *Gompf et al., 2017*).

Band-pass filtered low beta band signals in the SMA and auditory association cortex were normalized with the average over both conditions to reduce inter-subject-variability and plotted as a function of time during the sequence of four auditory beats, separately for the slow and fast tapping condition. To detect an amplitude modulation of the low beta band signal (*Morillon and Baillet, 2017*; *Fujioka et al., 2015*) at a temporal modulation rate that corresponded to the fast auditory beat rate (2.5 Hz), a fast Fourier transformation was applied to the normalized band-pass signals (resulting frequency resolution: 0.5 Hz). Power spectra of each source were z-transformed for every participant. PSD values of low-beta band modulation at auditory beat frequency (2.5 ±0.5 Hz) were tested for differences between the left and the right auditory cortex as well as between the left and right primary hand motor cortex and between the primary handmotor cortices and the SMA, and between left and right cerebellum and the SMA using non-parametric permutation dependent t-tests (999 permutations, alpha = 0.05). Since sequences of four taps did not allow for analysis of a temporal modulation rate at 0.625 Hz, we additionally investigated entire trials for a temporal modulation of low beta power at the slow tapping rate (0.625 Hz). PSD values of low-beta power modulation at the slow tapping frequency (0.625 ±0.25 Hz) were tested for condition differences in the left and right auditory cortices as well as in the SMA (non-parametric permutation dependent t-tests, 999 permutations, alpha = 0.05). We additionally tested whether the representation of the slow tapping rate in the spectrum of the right auditory association cortex was stronger compared to the left auditory association cortex (non-parametric permutation dependent t-tests, 999 permutations, alpha = 0.05).

Single trial data of slow tapping are illustrated in *Figure 6*, lower panels and *Figure 7*, upper right panel. Averages of beta power sequences are plotted in *Figure 6*, upper panels, and *Figure 7*, upper left panel. The lower panels in *Figure 6* and the right upper panel in *Figure 7* illustrate the average beta power during two sequences of slow tapping for better illustration of the slow rhythm.

## Relationship between low beta power modulation and timing variability

To investigate the relationship between the low beta band activity and inter-tap-intervals (ITI), source data of the SMA, left auditory cortex and right auditory cortex was used. ITIs for the slow tapping condition ranged from 1384 to 1816 ms, for the fast tapping condition ITI range was from 268 to 568 ms. Sequence data in the slow tapping condition (four beats) were cut from the time point of maximum pressure - 500 ms and + 2500 ms to include also too long intervals. Low beta power over time was extracted and sequences were split into sequences with long ITIs (ITI > 66 th percentile), n = 517, medium ITIs (33 th percentile <ITI < 66 th percentile), n = 1567 and short ITIs (ITI < 33 th percentile), n = 517. The same procedure was performed for fast taps.

For investigating effects at the end of the sequence, the ITI range of 400 ms was used for comparison. To investigate effects at beat position one, the time of the mean beta peak in the medium ITI trials (500 ms after the tap for slow and 220 ms after the tap for fast) was used as the center point for a second window spanning also 400 ms. Long and short ITI sequences were compared in these two windows using a non-parametric independent permutation t-test with cluster correction (alpha at 0.5).

## TPDC

Using a time-frequency causality method allows analyzing the temporal dynamics of causality with frequency resolution (*Figure 2* step 4). The TPDC is based on dual-extended Kalman filtering (*Wan and Merwe, 2002*), and allows time-dependent auto-regressive coefficients to be estimated, independent of the underlying frequency power in the timeseries. At each time point, previous state and weight estimates were fed to both the Kalman filters. Both predictors were then corrected on the basis of observed data such that they yield current state and weight estimates. By using two Kalman filters working in parallel with one another, both states and model parameters of the system were estimated at each time point. The time-dependent multivariate autoregression (MVAR) coefficients were used to calculate the causality between the time series. By calculating the time-dependent MVAR coefficients at each time point, partial-directed coherence (PDC; *Blinowska, 2011*),

based on the principle of Granger causality, was computed. Based on the fMRI results with increased activity in the right but not the left auditory association cortex and the SMA for slow compared to fast tapping, the Fourier transform of the MVAR coefficients and PDC was calculated between the SMA and the left and right auditory association cortices, respectively, resulting in two auditory-motor and two motor to auditory connections. TPDC was calculated for every sequence individually. Afterwards, slow and fast sequences were averaged separately. This resulted in 2000 time points and 161 frequency bins (1–80 Hz). The first 200 time points were not used for further analyses due to the Kalman filter adaptation period.

Values of directed connectivity were validated with a reverse technique that tests for effects of volume conduction (*Haufe et al., 2013*). We compared original connectivity with reversed connectivity. This analysis did not show significant differences between original and reversed connectivity in a permutation t-test (p>0.05), which excludes volume conduction effects.

Like the analyses of the source signals, further connectivity analyses were also focused on the low-beta band. Therefore, the time-frequency representation of the connectivity analysis was reduced by extracting the average over frequencies in the low-beta band. The band pass filtered signal was smoothed with an average mean filter of 100 ms windows with 10 ms steps.

Connectivity results were tested on power differences in the low beta band and focused again on the difference between slow and fast tapping. For power differences, contrasts between slow and fast tapping were tested in a two factorial non-parametric permutation ANOVA with factors hemisphere (left and right) and direction (auditory to motor and motor to auditory, with motor cortex being represented by the SMA). Alpha was set to 0.05.

To detect temporal modulations of low beta TPDC, the band-pass filtered signal was transformed into Fourier space. Resulting power spectra were normalized by subtracting the axis distance for every subject, and z-transformed. Frequency peaks in the overall 1/f distribution were validated following (*Haller et al., 2018*). The algorithm uses automatic parameterization of neural power spectral densities as a combination of the aperiodic signal and putative periodic oscillations with no a priori specification of band limits.

To investigate the relationship between directed connectivity and behavioral parameters, timing variability in the slow tapping condition was correlated using Pearson's correlation coefficient with the significant connectivity contrasts (slow >fast). Alpha was set to 0.05.

## Acknowledgements

This study was funded by the German Research Foundation with an Emmy Noether Grant to CAK (KE 1514/2–1) and by a LOEWE grant of the state of Hesse (NEFF). SG received funding from the German Research Foundation (SFB CRC-128 and SFB CRC-1193) and MM and SG from the Boehringer Ingelheim Fonds BIF-03. We are thankful to Alla Brodski and Georg Friedrich Paatsch for assisting the MEG measurements.

## Additional information

### Funding

| Funder | Grant reference number | Author |
|---|---|---|
| Deutsche Forschungsgemeinschaft | KE 1514/2-1 | Christian Alexander Kell |
| State of Hesse | LOEWE grant NEFF | Christian Alexander Kell |
| Deutsche Forschungsgemeinschaft | SFB CRC-128 | Sergiu Groppa |
| Deutsche Forschungsgemeinschaft | SFB CRC-1193 | Sergiu Groppa |
| Boehringer Ingelheim Fonds | BIF-03 | Sergiu Groppa Muthuraman Muthuraman |

The funders had no role in study design, data collection and interpretation, or the decision to submit the work for publication.

## Author contributions
Anja Pflug, Florian Gompf, Data curation, Formal analysis, Investigation, Visualization, Methodology, Writing—original draft; Muthuraman Muthuraman, Software, Formal analysis, Methodology, Writing—review and editing; Sergiu Groppa, Conceptualization, Writing—review and editing; Christian Alexander Kell, Conceptualization, Resources, Supervision, Funding acquisition, Writing—original draft, Project administration, Writing—review and editing

## Author ORCIDs
Muthuraman Muthuraman (iD) http://orcid.org/0000-0001-6158-2663
Christian Alexander Kell (iD) https://orcid.org/0000-0002-6299-0076

## Ethics
Human subjects: Experimental procedures were approved by the ethics committee of the medical faculty of Goethe University (GZ 12/14) and are in accordance with the declaration of Helsinki. All participants gave written informed consent.

## Decision letter and Author response
Decision letter https://doi.org/10.7554/eLife.48404.022
Author response https://doi.org/10.7554/eLife.48404.023

## Additional files

### Supplementary files
• Transparent reporting form DOI: https://doi.org/10.7554/eLife.48404.018

### Data availability
All data are available for download at https://doi.org/10.5061/dryad.pg4f4qrj6.

The following dataset was generated:

| Author(s) | Year | Dataset title | Dataset URL | Database and Identifier |
|---|---|---|---|---|
| Pflug A, Gompf F, Muthuraman M, Groppa S, Kell CA | 2019 | Data from: Differential contributions of the two human cerebral hemispheres to action timing | https://doi.org/10.5061/dryad.pg4f4qrj6 | Dryad Digital Repository, 10.5061/dryad.pg4f4qrj6 |

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
