## [Decision Letter]

Thank you for submitting your article "Differential contributions of the two human cerebral hemispheres to action timing" for consideration by *eLife*. Your article has been reviewed by two peer reviewers, one of whom is a member of our board of Reviewing Editors, and the evaluation has been overseen by Richard Ivry as the Senior Editor. The following individual involved in review of your submission has agreed to reveal their identity: Hugo Merchant (Reviewer #1).

The reviewers have discussed the reviews with one another and the Reviewing Editor has drafted this decision to help you prepare a revised submission.

The reviewers found this study interesting in addressing the brain basis for the previously-described phenomenon that the right hand is more precise at rapid tapping and the left at slower tapping. This might have a brain basis linked to different sensory processing of different rates, different motor precision or different sensorimotor integration. They liked the study design in which different rates of motor output were linked to an identical sensory input.

Essential revisions:

1) A main concern is that the MEG results depend on grouping all trials. This study would benefit from exploring single trial effects, especially comparing the magnitude of low beta oscillations during correct and incorrect (large performance error) trials for both the slow and fast tapping tempos (see Cadena-Valencia et al., 2018, *eLife*).

2) The beta modulation in response to fast tapping that was greater in left association cortex in Figure 5 convincing but we were not so sure about the right modulation in response to slow tapping which seems a little off the beat and would like to have seen over more than one cycle of four sounds.

3) Why did the authors not use the fMRI results directly as an ROI for the beamformer analysis?

4) The beamforming seems to have worked quite well (can be problematic for highly correlated symmetrical sources) but this is not a great technique for assessing activity in the putamen and cerebellum which might also be relevant, and appear to have been demonstrated in the fMRI results consistent with a number of previous studies. We would be interested to hear the authors' comments on any synchronisation to tapping in those areas: were these examined as well as the motor cortex, or was this technically not feasible?

5) We found the examination of auditory motor interactions based on functional connectivity interesting and plausible. In the event basal ganglia and cerebellum could be assessed, those could have been included in the analysis.

6) In terms of connectivity the analysis reasonably focuses on functional connectivity between areas (e.g. auditory cortex and SMA) using coherence. Given that hierarchal models are addressed, was there any attempt to examine effective connectivity to assess directional influences (e.g. Granger)?

---

## [Author Response]

Essential revisions:1) A main concern is that the MEG results depend on grouping all trials. This study would benefit from exploring single trial effects, especially comparing the magnitude of low beta oscillations during correct and incorrect (large performance error) trials for both the slow and fast tapping tempos (see Cadena-Valencia et al., 2018, eLife).

We thank the reviewers for this excellent idea. We consequently investigated whether timing errors (too short vs. too long inter-tap-intervals) were associated with differences in low beta power modulation. Too long intervals during slow tapping, the primary condition of interest in this study, were associated with an even larger beta power modulation in right, but not left auditory association cortex, arguing further for a behaviorally relevant lateralized representation of rhythms. While timing variability during slow tapping related to beta amplitude effects at auditory beat position 1 in the right auditory cortex, the SMA showed only a relationship between timing variability and the duration of beta suppression around auditory beat 4, with a latency coinciding with the actual tap. It was only during fast tapping at 2.5Hz that the SMA showed amplitude effects of timing variability, arguing in favor of an active role of the SMA in timing motor output in the subsecond range. Please see the new Results subsection “Low beta band power modulations explain timing variability”, the respective paragraph in the Discussion subsection “Rhythm representations in the auditory association cortices” and the new Materials and methods subsection “Relationship between low beta power modulation and timing variability”.

2) The beta modulation in response to fast tapping that was greater in left association cortex in Figure 5 convincing but we were not so sure about the right modulation in response to slow tapping which seems a little off the beat and would like to have seen over more than one cycle of four sounds.

We added illustrations of two cycles of low beta power modulations during slow tapping to the new Figures 6 and 7 showing that the slow power modulation indeed cycles with maxima around beat position 1 which corresponds to the time point of the beta rebound after the tap at auditory beat position 4. These new panels illustrate nicely a stronger representation of this slow rhythm in the SMA and the right auditory association cortex with a maximal beta suppression at the tap at beat position 4 and a weaker representation in the left auditory association cortex. Please note that the power modulation at the slow tapping rate is detected in the power spectrum in the entire trials and also visible in single trial data, which we added to the revised figures.

3) Why did the authors not use the fMRI results directly as an ROI for the beamformer analysis?

We knew from previous work, especially by Pollok and colleagues that cortico-muscular coherence represents an excellent measure to source-localize finger-tapping related MEG activity. Because not all participants of the fMRI experiment also underwent MEG scanning, individual MRI coordinates were only available for a subgroup of participants. Also, there is active discussion whether MEG sources and BOLD activations reflect comparable processes. After a first publication wave of good fits between fMRI and MEG source localization results (which aimed to prove the “quality” of MEG source localizing techniques), especially Riitta Hari and Riitta Salmelin nowadays argue for a superior sensitivity of MEG for feedforward signals and of fMRI for feedback signals (see, e.g., Vartiainen et al., Journal of Neuroscience, 2011; Renvall et al., Cerebral Cortex, 2012; Kujala et al., NeuroImage, 2014).

4) The beamforming seems to have worked quite well (can be problematic for highly correlated symmetrical sources) but this is not a great technique for assessing activity in the putamen and cerebellum which might also be relevant, and appear to have been demonstrated in the fMRI results consistent with a number of previous studies. We would be interested to hear the authors' comments on any synchronisation to tapping in those areas: were these examined as well as the motor cortex, or was this technically not feasible?

Beamforming based on cortico-muscular coherence also revealed a left and a right cerebellar source, which we now include in our study. The results show that low beta power modulation in these cerebellar sources, alike in the motor cortex, only mirrors the actual motor output. This does not exclude a cerebellar contribution to internal timing, however our analyses did not reveal such an effect. Because neither putaminal nor thalamic sources were detectable in every single participant, we did not include these regions in our manuscript. However, we added the following sentences to the Discussion, emphasizing the contribution of subcortical brain regions to rhythm processing: “Low beta power in the cerebellar sources did only mirror the motor output in our experiment. […] We cannot rule out that other cerebellar sources with less cortico-muscular coherence compared to the cerebellar sources identified here show internal timing-related profiles”.

5) We found the examination of auditory motor interactions based on functional connectivity interesting and plausible. In the event basal ganglia and cerebellum could be assessed, those could have been included in the analysis.

Because putaminal sources were only found in some of our participants (see above), we could not include the basal ganglia into our connectivity matrix. Since cerebello-cortical effective connectivity is mediated by the thalamus (Gross et al., 2002; Pollock et al., 2005) and thalamic sources could not be detected in every participant, we also could not add these connections to our model. We nevertheless acknowledge the contribution of subcortical regions to rhythm processing in the Discussion, see also the answer to comment number 5.

6) In terms of connectivity the analysis reasonable focuses on functional connectivity between areas (e.g. auditory cortex and SMA) using coherence. Given that hierarchal models are addressed, was there any attempt to examine effective connectivity to assess directional influences (e.g. Granger)?

The connectivity measure used here actually is a measure of effective connectivity. Time-resolved partial directed coherence (TPDC) has been applied to EEG, MEG or fMRI signals (Anwar et al., Brain Topography, 2016). This method is based on directed effective connectivity to compute a time-frequency analysis established on an MVAR model. Allowing the application over time of multiple PDC (Baccalá and Sameshima, Biological Cybernetics, 2001), TPDC makes it possible to account for the evolutions over time and frequency bands of the information transfer directed between multiple time series, i.e. between regions of interest.